# Integrative Computational Approaches for Understanding Drug Resistance in HIV-1 Protease Subtype C

**DOI:** 10.3390/v17060850

**Published:** 2025-06-16

**Authors:** Sankaran Venkatachalam, Nisha Muralidharan, Ramesh Pandian, Yasien Sayed, M. Michael Gromiha

**Affiliations:** 1Department of Biotechnology, Bhupat and Jyoti Mehta School of Biosciences, Indian Institute of Technology Madras, Chennai 600036, India; bt23d078@smail.iitm.ac.in (S.V.); bt24d025@smail.iitm.ac.in (N.M.); 2Protein Structure-Function Research Laboratory, School of Molecular and Cell Biology, University of the Witwatersrand, Johannesburg 2000, South Africa; ramesh.pandian@wits.ac.za (R.P.); yasien.sayed@wits.ac.za (Y.S.)

**Keywords:** HIV protease, subtype C, computational studies, MD simulations, AIDS, mutation, insertion, structure activity relationship

## Abstract

Acquired immunodeficiency syndrome (AIDS) is a chronic disease condition caused by the human immunodeficiency virus (HIV). The widespread availability of highly active antiretroviral therapies has helped to control HIV. There are ten FDA-approved protease inhibitors (PIs) that are used as part of antiretroviral therapies in HIV treatment. Importantly, all these drugs are designed and developed against the protease (PR) from HIV subtype B. On the other hand, HIV-1 PR subtype C, which is the most dominant strain in countries including South Africa and India, has shown resistance to PIs due to its genetic diversity and varied mutations. The emergence of resistance is concerning because the virus continues to replicate despite treatment; hence, it is necessary to develop drugs specifically against subtype C. This review focuses on the origin, genetic diversity, and mutations associated with HIV-1 PR subtype C. Furthermore, computational studies performed on HIV-1 PR subtype C and mutations associated with its resistance to PIs are highlighted. Moreover, potential research gaps and future directions in the study of HIV-1 PR subtype C are discussed.

## 1. Introduction

Human immunodeficiency virus (HIV) is the causative agent of acquired immunodeficiency syndrome (AIDS), one of the most deadly diseases in human history [1,2]. According to the 2023 Global HIV and AIDS statistics factsheet from UNAIDS, it is estimated that a total of 39.9 million people are living with HIV, and about 630, 000 people died due to AIDS-related illnesses [3]. HIV belongs to the retrovirus family containing ribonucleic acid (RNA) as its genetic material [4]. It targets the host immune system, causing morbidity and mortality. The entry of HIV into the host cell is facilitated by its interaction with CD4+ lymphocytes and coreceptors (CXCR4 or CCR5), allowing the virus to inject its genetic material into the host cell [5]. Upon entry, the viral RNA is converted to viral deoxyribonucleic acid (DNA). Subsequently, the converted DNA integrates with the host genomic DNA, where it undergoes transcription and translation to yield viral proteins [6].

The replication of HIV is critically dependent on a set of key enzymes that enable this life cycle. Three major enzymes are critical to the life cycle of HIV: (i) reverse transcriptase (RT), which converts the viral RNA to DNA; (ii) integrase (IN), which integrates the converted DNA to the host DNA; and (iii) protease (PR), which is essential for cleaving several polypeptides (Gag, Gag-Pol and Nef) required for viral maturation. Most of the HIV drug design studies are focused on inhibiting these enzymes [6]. Zidovudine (ZDV), a nucleoside reverse transcriptase inhibitor, was the first approved drug for treating HIV infections [6]. In the subsequent years, drugs targeting PR, IN, and RT were developed to impede viral replication within the host [7,8,9]. These drugs rendered a significant HIV treatment strategy when used in combination; however, they were not 100% effective [10,11]. Despite the advancements over the past four decades in the fields of medicine, enzymology, and computer-aided drug design, complete eradication of the virus from the human host remains a formidable challenge. A significant barrier to effective drug design is the genetic diversity in HIV, which is the result of variations that occur during reverse transcription with the dual mechanism of retroviral recombination and mutations owing to the emergence of different subtypes [12,13,14]. The emergence of subtypes is associated with drug resistance mechanisms and has implications for drug development [15].

Among these subtypes, understanding the distribution and prevalence of HIV-1 is essential for effective treatment. Infections resulting from HIV can be attributed to either HIV-1 or HIV-2, with the former being predominant while the latter is uncommon and localized [16]. Furthermore, the genetic diversity within HIV-1 has resulted in many groups and subtypes. The group M (major group) accounts for 70% of the infections worldwide, with nine distinct subtypes: A, B, C, D, F, G, H, J and K. The other groups include group N (new), group O (outlier), and group P [17]. Notably, the distribution of these subtypes is uneven across the globe. For instance, subtype B is predominantly observed in America, Western Europe, and Australia [18,19,20], whereas subtype C is predominant in Southern Africa, Ethiopia, and Southeast Asia (India) [20]. Until recently, the subtype B strain was observed to be the predominant causative for HIV infections. However, recent data indicates a shift in the trend and that subtype C now accounts for ~46% of HIV infections worldwide [20,21]. This shifting pattern in global prevalence underscores the importance of investigating subtype C.

Twenty years after the first instance of HIV-1 (1930s), the subtype C is thought to have originated in Mbuji-Mayi in the Democratic Republic of Congo in the 1950s. Subsequently, infections spread from Mbuji-Mayi to South Africa through its bordering countries, including Lubumbashi, Zimbabwe, Uganda, Kenya, Ethiopia, and Tanzania. Factors such as increased migration, socio-political scenario, and increased trade facilitated the dissemination of subtype C variants in South Africa [5,21]. Figure 1 describes the origin, timeline, and spread of subtype C. Tracing the origin and spread of subtype C helps in understanding its current global dominance. Despite accounting for nearly half of the infections worldwide [20,21], subtype C remains less explored than the subtype B variant. Additionally, the inhibitors developed to target the key enzymes such as RT, PR, and IN are primarily designed for subtype B [22], rendering them ineffective when used for subtype C [23]. RT and IN play vital roles in various stages, including RNA to DNA conversion and viral-host DNA integration, while PR is crucial for the maturation of the virus [24].

Among the viral enzymes, the PR remains one of the most extensively studied targets for mitigating HIV replication owing to its functional importance. This resulted in the development of ten Food and Drug Administration (FDA)-approved drugs such as (i) saquinavir (SQV), (ii) indinavir (IDV), (iii) ritonavir (RTV), (iv) nelfinavir (NFV), (v) amprenavir (APV), (vi) fosamprenavir (FPV), (vii) lopinavir (LPV), (viii) atazanavir (ATV), (ix) tipranavir (TPV) and (x) darunavir (DRV). Notably, all the approved inhibitors were designed to target subtype B PRs for various reasons, as discussed below [25,26,27]. Protease inhibitors (PIs) have played a pivotal role in the treatment of HIV, particularly during the late 1990s and early 2000s in regions such as the United States and Europe, where subtype B was predominant. Although combination therapies involving PIs with reverse transcriptase inhibitors were effective, the development of resistance mutations toward PIs had unfavorable patient outcomes. This led to the development of novel therapies, such as VOCABRIA, DOVATO, and CABENUVA, that do not incorporate PIs [28,29]. While PRs remain one of the critical targets for mitigating HIV replication, the development of new PIs has been impaired in recent years. The primary challenge lies in designing PIs that can effectively overcome multiple resistance mutations. Despite these challenges, the higher prevalence of subtype C compared to other subtypes, together with the poor efficacy of existing drugs, demands the development of PIs specific to subtype C [21,24].

Hence, a critical understanding of the mechanisms underlying drug resistance is required for subtype C PR. This review highlights various studies on HIV-1 PR subtype C, such as diversity, polymorphisms, mutations, and computational studies of specific drugs. Furthermore, it highlights the existing research gaps and future perspectives to study HIV-1 PR subtype C.

## 2. HIV Protease

### 2.1. Structure and Mechanism of HIV Protease

HIV PR is a functional homodimer comprising 99 amino acids in each subunit [30,31]. The three-dimensional structure of HIV PR is shown in Figure 2. The catalytic triad contains conserved active site residues Asp25, Thr26, and Gly27. It hydrolyses the peptide bond in the substrate gag-pol protein, resulting in the production of structural proteins such as matrix (MA), capsid (CA), nucleocapsid (NC), and functional enzymes (PR, RT, and IN) that constitute the core of the HIV. The catalytic mechanism of PR is described elsewhere [32]. Conformational changes that occur between the open and closed states of the PR, governed by the concerted movement of flaps, hinge, cantilever, and the fulcrum, are crucial for its function. The open state facilitates substrate/ligand binding and product release, whereas the closed state is crucial for substrate cleavage, yielding structural proteins [33,34,35]. Inhibitors are designed to impede the function of PR by competitively binding in place of the substrate, thereby attenuating the formation of structural proteins [36].

### 2.2. HIV-1 Protease Subtype C

The analysis of PR genes from drug-naive patients revealed that the HIV-1 PR subtype C sequence differed from the consensus subtype B at eight residues, namely, T12S, I15V, L19I, M36I, R41K, H69K, L89M, and I93L (Figure 3A) [39]. Figure 3A illustrates the sequence alignment between HIV-1 PR subtypes B and C, and Figure 3B highlights the locations of polymorphisms (S12, V15, I19, I36, K41, K69, M89, and L93) within the structure of HIV-1 PR subtype C. Although analogous substitutions are also present in subtype B PR sequences, the frequency of occurrence of these substitutions was notably higher in the HIV-1 PR subtype C. For instance, R41K was found in 100% of the subtype C sequences, whereas the frequencies for H69K, I93L, and M36I were 98.7%, 96.2%, and 87.3%, respectively. These findings have led to the classification of these substitutions as naturally occurring polymorphisms corresponding to subtype C [39,40]. These polymorphisms are naturally occurring variations in the PR, whereas drug-resistant mutations are induced by PIs. The emergence of polymorphisms did not alter the mechanism of the PR. However, it diminishes the efficacy of the PIs, posing a formidable challenge.

Coman et al. [41] elucidated the structure of HIV-1 PR subtype C complexed with NFV. They engineered 14 different variants of subtype B and subtype C to study the changes underlying these polymorphisms. The structural analysis revealed that these polymorphisms are primarily located in the functionally critical regions of the enzyme, including the hinge (M36I), fulcrum (T12S, I15V, and L19I) cantilever (H69L), and base (L89M and I93L). Furthermore, the study concluded that the polymorphisms, combined with the drug-resistant mutations, may lead to evasion and reduced drug response, ultimately leading to treatment failure.

## 3. Mutations in the HIV-1 Protease Subtype C

Mutations in HIV PR arise from various factors, including genetic diversity, selection pressure, and prolonged exposure to antiretroviral drugs [42]. These mutations in the HIV PR are categorized into two groups: (i) primary mutations or major mutations, which occur in the active site, and (ii) secondary mutations or minor mutations, which occur distal to the active site. The former has a direct effect on the drug binding and efficacy, while the latter arises to support the primary mutation [43,44]. Ledwaba et al. [40] reported that approximately 32% of the amino acid residues in HIV-1 PR subtype C undergo mutations. It is noteworthy that the majority of reported mutations for HIV PR are substitutions, with insertions and deletions occurring infrequently.

### 3.1. Substitutions in HIV-1 Protease Subtype C

Substitution refers to the replacement of one amino acid by another. Kantor et al. [25] reported three new substitutions, namely, K20T, K20I, and K20M, in addition to K20R in HIV PR, which emerged after the antiretroviral treatment (ART). Furthermore, substitutions in positions 50, 73, and 84 were observed post-ART. Similarly, the sequence comparison between ART-naive and treated patients (an LPV/RTV-containing regimen) showed that for patients who have substitutions L33F, I54V, and V82A, the therapy was not effective [41]. These drug-induced substitutions pose a formidable challenge in designing novel drugs for inhibiting HIV PR.

Apart from the drug-induced substitutions, some of the commonly observed substitutions in HIV-1 PR subtype C include M36I/V, L63P/T, and V82I. These substitutions are associated with low susceptibility to PR inhibitors. Specifically, the V82I mutation demonstrated low susceptibility to ATV [41]. Moreover, a study on drug-resistant substitutions in the HIV-1 PR subtype C gene indicated that minor substitutions corresponding to ATV and TPV were present in 100% of the ART non-responsive patients. Furthermore, patients who were non-responsive to ART harbored two major substitutions, D30N and M36I [45]. Additionally, the minor substitutions at positions 36, 63, 69, and 93 were prevalent among most ART unresponsive patients. A comprehensive list of drug-resistant substitutions in HIV-1 PR subtype C is provided in Table 1, and their respective locations on the HIV-1 PR subtype C structure are shown in Figure 4.

In addition to drug-resistant substitutions, some other substitutions were also reported in the literature (Table 2). Although the role of these substitutions is not clear, they are believed to complement the drug-resistant substitutions in decreasing drug susceptibility.

Furthermore, residues such as I50, A/V28, D30, and V82 (Figure 4) spanning the substrate groove of HIV are reported to be critical for interaction with the PIs [46,47]. Thus, substitutions such as V82I, V28A, and D30N (Table 1 and Table 2) will impair the interaction of PI with the HIV PR subtype C, which may affect its binding and lead to drug resistance.

### 3.2. Insertions in HIV-1 Protease Subtype C

An amino acid insertion represents a distinct category of mutation, which is rarely observed in HIV PR. In HIV PR subtype B, insertions typically occur between residues 32 and 41 within/close to the hinge region. Similarly, in HIV-1 PR subtype C, insertions predominantly occur between residues 35 and 39 (hinge region). The observed prevalence of insertions at positions E/D35, I36, and L38 are 0.22%, 0.13%, and 0.13%, respectively [40]. The list of insertions reported for HIV-1 PR subtype C is presented in Table 3. These insertions have been shown to increase the sensitivity of the virus to antiretroviral drugs such as LPV, ATV, and DRV, thereby reducing drug susceptibility. It is important to note that these insertions occur in conjunction with substitutions. Additionally, certain insertions such as I36↑T↑T were found to be present in PI drug-naive patients treated with RT inhibitors (RTIs) such as Efavirenz, d4t (Stavudine) and 3TC (Lamivudine) [48].

Mutations in HIV-1 PR subtype C induce structural changes. Such alterations directly affect the binding of a drug to HIV-1 PR subtype C, enabling the virus to evade the effect of PIs. The binding free energy for drug molecules interacting with the PR is determined by kinetic and thermodynamic experiments. Mosebi et al. [23] performed kinetic and thermodynamic studies to compare the binding of FDA-approved inhibitors SQV, RTV, IDV, and NFV to subtypes B and C. The binding free energy (ΔG) values are summarized in Table 4. While differences between subtypes B and C are subtle, it was shown that the presence of mutations decreases the interaction strength of the drugs. For example, the change in binding free energy between mutant (E35D↑G↑S) and wild-type HIV PR subtype C PR complexed with ATV was reported to be 4.6  ±  0.2 kcal/mol, culminating in drug resistance [24,48,49].

## 4. Computational Studies on HIV-1 Protease Subtype C

Many extensive anti-HIV drug discovery studies were focused on HIV-1 subtype B PR through FDA-approved PIs [50,51]. On the other hand, these inhibitors generally show altered biological activities against subtype C PR [15]. Several experimental studies highlight the key differences in dynamic behaviors between the subtype B and C PR. Some of the advancements in computational biology offer detailed insights into molecular interactions, which are largely controlled by dynamics. Specifically, molecular dynamics (MD) simulations play a major role in understanding and visualizing the spontaneous opening of PR flaps, which has been validated [52]. Furthermore, MD simulations have been successfully used to gain detailed molecular-level insights into the conformational dynamics of HIV-1 PR variants [53,54].

Naicker et al. [55] performed a molecular dynamics study to understand the impact of a specific point substitution at position 36 in HIV PR subtype C on the stability of the hinge region, which in turn affects the flap flexibility. The study reported that the E35-R57 salt bridge is absent in both chains of the HIV-1 PR subtype C, as R57 adopts a different rotamer when compared to the subtype B PR. In HIV-1 PR subtype C, R57 forms backbone hydrogen bonds with V77 in a region that is located interior of the flaps. The absence of the E35-R57 salt bridge leads to a clear outward movement of the flaps. Any disruption of the salt bridge in the hinge region is shown to reduce the stability of HIV-1 PR subtype C, which in turn contributes to high resistance to PIs. Recently, Mishra et al. [56] explored the interactions between the HIV-1 PR subtype C and CD4 through molecular docking and MD simulations. CD4 (Cluster of Differentiation 4) is a surface receptor found primarily on T-helper lymphocytes, which play a central role in coordinating immune responses. HIV targets and binds to CD4 through its envelope glycoprotein, initiating the entry process. This interaction is the first critical step in viral infection, allowing the virus to attach to the host cell before engaging a coreceptor (typically CCR5 or CXCR4) and fusing with the cell membrane [57]. The study highlighted some of the key interacting residues, such as Asp25, Gly27, Ile50, Gly51, Pro81, and Ile84 in HIV PR, which can be used for novel drug development.

### 4.1. Computational Studies on First-Generation HIV Protease Inhibitors Against HIV-1 Protease Subtype C

Lockhat et al. [48] studied the structural characteristics of NFV through MD simulations to understand its weak binding to the mutant I36T↑T HIV PR. It was observed that NFV induced a semi-open conformation of the PR flaps that led to increased inhibitor mobility and reduced binding to the PR. This altered flap dynamics was attributed to substitutions and insertions within the hinge region that governs flap movement and distinguishes the I36T↑T mutant from the wild-type. Moreover, a significant reduction in hydrogen bonds and hydrophobic interactions with catalytic residues was identified. This further explains the poor binding affinity of NFV to the HIV-1 PR subtype C. These findings collectively suggest that the I36T↑T insertion disrupts the structural integrity and interaction ability of the PR with NFV. The MD simulations study and the binding free energy calculations performed by Ahmed et al. [24] revealed that most inhibitors exhibit slightly lower activity towards subtype C when compared to subtype B, and they reported that this could be most likely due to the indirect effect of eight polymorphisms present in subtype C which influence flap motion and enhance the flexibility of HIV-1 PR subtype C. However, among the first-generation inhibitors, RTV and APV showed better binding affinities towards HIV-1 PR subtype C due to electrostatic and hydrogen bonding interactions of the inhibitors with the subtype C PR.

Furthermore, Ahmed et al. [58] revealed that substitutions V82A and V82F/I84V within the HIV PR induce significant alterations in the binding profiles of RTV, SQV, IDV, and NFV, influencing crucial flap residues. This binding free energy study was performed using the MM/GBSA method. While RTV experiences both favorable and unfavorable binding changes, SQV and IDV exhibit a general decrease in binding affinity, with IDV showing particular sensitivity at the flap tips. The reduced binding of NFV with the PR is primarily because of the disruption in the interactions rather than direct steric clashes. In addition, disrupted interactions at Ile84/Ile84′ contributed to the reduced binding of NFV. These observations suggested that there is a substitution-induced distortion in the geometry of the binding site instead of a change in its chemical properties. This highlights the complex interplay between active site mutations and drug-binding efficacy.

The activity of HIV PR is majorly dependent on the correlated movements of opening and closing of the flap region, which is essential for the enzyme function [51]. The computational analysis performed by Venkatachalam et al. [59] revealed that the L38HL mutation in HIV-1 PR induces structural alterations that contribute to SQV resistance. This specific mutation widens the active site of the PR, leading to the disruption of critical interactions of residues Asp25′, Ala28′, and Ile82′ with SQV, thereby reducing drug binding affinity. This structural change, along with increased flexibility in the hinge and flap regions due to altered hydrogen bond interactions (Figure 5), impacts the dynamic behavior of the PR and its correlated residue motion. Overall, the effect of these changes leads to a less favorable conformation for SQV binding owing to diminished drug efficacy and increased drug resistance.

Sanusi et al. [60] studied the binding free energies of HIV-1 PR inhibitors towards HIV-1 PR subtype C using a two-layered Own N-layered Integrated molecular Orbital and molecular Mechanics (ONIOM) model. They treated the inhibitors and catalytic active residues (ASP 25/25′) with high-level quantum mechanics (QM) theory and AMBER force field for the remaining residues. They reported that the hydrogen bond distance for NFV bound to the L38L↑N↑L PR is longer than that of SQV. This increased distance corresponds to a weaker theoretical binding free energy of NFV.

Overall, the computational studies involving first-generation antiretroviral drugs were found to possess less favorable interactions with HIV-1 PR subtype C compared to subtype B PR.

### 4.2. Computational Studies on Second-Generation HIV Protease Inhibitors Against HIV-1 Protease Subtype C

Ahmed et al. [24] reported an effective comparative analysis of nine FDA-approved drugs, such as RTV, APV, SQV, IDV, NFV, LPV, DRV, TPV, and ATV, against PR subtype B and HIV-1 PR subtype C using MD simulations and binding free energy calculations. This computational methodology was assessed using experimental binding free energies as a benchmark for subtype B. The calculated results for second-generation drugs such as LPV and DRV show better binding affinities towards HIV-1 PR subtype C when compared with other inhibitors. Interestingly, these inhibitors exhibit strong interactions with the flap residues, resulting in an increased binding affinity. The interactions of DRV with Asp29 and Asp30 of the PR form a conserved hydrogen bond between the oxygen atoms of the bis-tetrahydrofuranyl moiety. This bonding, coupled with favorable van der Waals forces, plays a major role in enhanced binding affinity. Conversely, the study on comparative analysis of DRV bound to HIV-1 PR subtype C and its insertion variant L38HL showed a decreased affinity for the DRV-L38HL complex [61].

Halder et al. and Honarparvar [62] analyzed the dynamics and interactions of the drugs ATV and DRV towards both subtypes B and CPR. They showed altered responses in both PR subtypes with these drugs using different post-dynamic analyses such as cross-correlation, principal component analysis (PCA), per-residue decomposition, and hydrogen bond analyses. Interestingly, they reported that the binding behavior of these inhibitors determined the overall binding energy rather than the PR subtype. For instance, I50L substitution lowered the binding energies of DRV with subtype B PR, whereas it increased the binding affinity of ATV with subtype B PR [63]. The analysis on residual decomposition energy suggested that DRV interactions are mediated by van der Waals and electrostatic interactions, whereas ATV interactions are mainly through hydrophobic interactions. The replacement of Ile50 with other hydrophobic amino acids such as valine or leucine affected the hydrophobic network with ATV [62].

Sankaran et al. [37] reported that the insertion in the L38HL variant of HIV-1 PR increases the flexibility at the hinge regions (Figure 6), leading to a unique binding mechanism for ATV by creating intramolecular hydrogen bonding in the ligand. This restricted the conformational flexibility and promoted a compact structure of the ligand. This structural constraint, along with the increase in the active site volume due to flap dynamics, leads to enhanced ATV binding affinity. Specifically, the observed entropy reduction in ATV, resulting from its intramolecular bonding, is compensated by an enthalpic gain through increased hydrophobic and van der Waals interactions within the slightly expanded active site. Furthermore, ATV displayed a shift in the intermolecular hydrogen bonding network from active site residues in the wild-type to flap residues in L38HL, thereby facilitating local conformational changes that stabilized the complex. This study demonstrated that hinge region insertions not only alter the overall dynamics of the PR but also directly influence ligand conformation and binding interactions, potentially leading to differential drug susceptibility. This is evidenced by the increased affinity of ATV for the L38HL variant when compared to the wild-type PR.

Several other mutations reported in HIV PR increase the drug resistance, resulting in reduced affinity of the currently FDA-approved drugs against the PR [64]. Sanusi et al. [65] employed the ONIOM method to understand the binding affinities of the nine HIV PIs against double insertion L38L↑N↑L PR. They observed that these mutations possessed a considerable effect on the binding affinities of PIs to the PR. On comparing the hydrogen bond distances of TPV and ATV with the mutant PR, it is revealed that TPV has stronger hydrogen bond interactions than ATV. Furthermore, they reported that explicitly including and appropriately treating water molecules in ONIOM modeling of HIV PR active site is crucial for accurately simulating catalytic interactions.

### 4.3. Computational Studies on Other Potential Inhibitors Against HIV-1 Protease Subtype C

Since no conventional drugs have been found to cure HIV/AIDS to date, Shode et al. [66] explored the inhibitory potential of certain bioactive compounds from South African indigenous plants against HIV-1 PR subtype C using molecular dynamics simulations. The binding free energies of phytochemicals such as cyanidin-3-glucoside and maslinic acid were found to be within the range of FDA-approved drugs such as LPV and DRV. This study also revealed that these metabolites interacted hydrophobically with the active site amino acid residues, which led to identifying other key residues involved in HIV PR inhibition. This could lead to the design of novel drugs.

## 5. Quantitative Structure-Activity Relationship: Protease Subtype B Versus C

Quantitative structure–activity relationship (QSAR) is a mathematical model which is used in understanding the factors that influence the activity and also in designing new potential compounds. Certain molecular descriptors, such as chemical features and topological indices, are used to characterize the structural features of molecules [67]. These features are further related to biological activities using various machine learning (ML) tools such as neural networks (NNs), multiple linear regression (MLR), genetic algorithms (GAs), and support vector machines (SVM). These computational models are useful in identifying key structural features that contribute to the strong binding affinity of HIV PR with the inhibitors. This information would help in designing new inhibitors with enhanced potency and selectivity.

Burbidge et al. [68] developed a 3D-QSAR model that identifies the key structural features of chalcone derivatives that are crucial for inhibiting HIV PR. Tong et al. [69] designed a 3D-QSAR model using a Topomer comparative molecular field analysis (CoMFA) search and predicted the activities of unknown compounds toward HIV PR subtype B. Zhang et al., [70] has established a 3D-QSAR model by incorporating hydrophobic and hydrogen bond donor fields into the CoMFA framework within the comparative molecular similarity indices analysis (CoMSIA). This approach enhanced the accuracy of the model and resulted in improved prediction ability and reliability. Similarly, Khedkar et al. [71] studied the electrostatic, steric, hydrophobic, and hydrogen-bonding properties of cyclic urea analogs and their impact on inhibitory activity using CoMFA and CoMSIA models. Bhargava et al. [72] designed a QSAR model for hydroxyethylamine derivatives as HIV PIs using Monte Carlo optimization. They showed that compounds containing stereochemical bonds, double bonds, and atoms such as nitrogen, oxygen, and sulfur exhibited higher inhibitory activity. Fatemi et al. [73] developed QSAR models based on docking-derived molecular descriptors to predict the inhibitory activities of 37 newly synthesized compounds against HIV PR subtype B. However, there is no QSAR model present for HIV-1 PR subtype C owing to the lack of a comprehensive dataset available for known HIV PIs targeting subtype C with experimentally measured inhibitory activity. By tailoring drugs/inhibitors specifically for HIV-1 PR subtype C, they could inhibit more effectively.

## 6. Unexplored Research Areas of HIV-1 Protease Subtype C

Panda et al. [74] studied nanomaterial-based ligands as they are more stable, resistant to many chemical reactions, and inflexible, thus maintaining the PR in an intact conformation. MD simulations and MM/PBSA free energy calculations were used to examine the interactions of pristine single-wall carbon nanotube (SWCNT) and functionalized SWCNT-OH carbon nanotubes with wild-type HIV-1 PR and primary mutant variants such as I50V, V82A, and I84V. They reported that the binding of SWCNT and SWCNT-OH into the active site was stable with a reduced flap open/close mechanism characterized by more effective binding than DRV. This further suggested that SWCNTs could be promising inhibitors and a better alternative to already available drugs. In conventional methods, inhibitors/drugs are designed to target the active site of the HIV PR. However, Badaya and Sasidhar [75] used a different approach of binding antibodies to the elbow region of the PR instead of the active site. This binding to the elbow region restricted movements in this region and further affected the flexibility of the flaps. The overall flexibility of the PR is affected due to the restricted movements of the flap region, which, in turn, restricts its catalytic activity. A similar approach can be used to design inhibitors targeting the elbow region for HIV-1 PR subtype C.

HIV Gag non-cleavage-site PI resistance mutations are another way in which HIV-1 can be resistant to PIs [76]. Parry et al. [77] showed that the presence of non-cleavage-site mutations in Gag may modulate PI resistance in subtype C viruses. For example, H219Q (in the capsid region), which was prevalent among treatment-naive HIV-1-infected individuals [78], and R409K (in the nucleocapsid region) have been identified in subtype C viruses from patients experiencing virological failure on PI-based regimens, despite the absence of major PR resistance mutations [77]. These mutations are believed to contribute to resistance by stabilizing the interaction between Gag and PR, referred to as forming a “substrate clamp”, thereby enhancing cleavage efficiency or modulating PI binding indirectly [78,79,80,81]. This mechanism represents a compelling target for future therapeutic strategies because the possibility of targeting Gag-PR interface interactions with novel inhibitors could lead to strategies that critically impair substrate recognition and restore PI susceptibility, especially for HIV-1 subtype C.

Tunc et al. [82] employed an ML model to construct drug isolate fold change (DIF) based artificial neural network (ANN) models to delineate the drug resistance potential of compounds that inhibit HIV-1 PR using genotype-phenotype data points of PIs from the Stanford HIV database (HIVDB) [83] (https://hivdb.stanford.edu/, accessed on 20 February 2025). To enhance the learning capacity of DIF models, Tunc et al. [84] employed a transfer learning approach using graph neural networks for the activity prediction of HIV PIs. This DIF model, in the presence of an isolate (i.e., a distinct viral variant defined by its unique amino acid sequence resulting from mutations, typically derived from patient samples), is effective in predicting drug resistance of novel PIs. Such ML-based models can be constructed to identify drug resistance in HIV-1 PR subtype C and to design novel inhibitors targeting subtype C.

Additionally, it has been reported that HIV-1 PR subtype B can bind additional residues outside of the cleavage site residues that are bound by the active site using the substrate grooves [46,47]. Studies linking substrate-groove residues to altered substrate binding in subtype C will be helpful in identifying additional residues that interact with the substrate, which can further be used to establish strong interaction inhibitors. Inhibitors designed using this approach can engage the secondary interaction sites and retain efficacy even in the presence of primary active-site resistance mutations, offering a beneficial strategy.

Furthermore, all these studies can be utilized for designing novel strategies to develop potent drugs targeting HIV-1 PR subtype C.

## 7. Future Perspectives

Several key areas of interest in computational research on HIV PR (specifically subtype C) are yet to be thoroughly explored. These areas include understanding the unique structural characteristics of HIV-1 PR subtype C, developing better inhibitors, and exploring the resistance of the enzyme to existing drugs. One such way of understanding is to explore the genetic variability within HIV subtype C, such as single-nucleotide polymorphisms (SNPs) or mutations, which could provide valuable insights into the response of different individuals to HIV treatment. Although few analyses on mutations have been studied, an extensive use of computational models can help to map all these variations and predict the impact of specific mutations impacting drug resistance. With the use of deep learning and ML, potential PIs can be predicted by analyzing large databases of chemical compounds and known inhibitors and by training models on PR-ligand interaction data [85,86,87]. This will help to identify compounds that are likely to inhibit HIV-1 PR subtype C.

The commercially available HIV PIs target the active site conventionally. However, there is a growing interest in targeting allosteric sites on the PR enzyme for novel drug designs that could overcome drug resistance mechanisms. Several computational approaches, such as QM, MD, and docking studies, can be used to identify potential allosteric sites and simulate their effects on HIV-1 PR subtype C. This would help in offering a better understanding of HIV PR behavior and interactions.

Drug repurposing is utilizing already available FDA-approved drugs for different diseases other than what they were originally developed for. This concept of drug repurposing is widely used nowadays for research to identify potential drugs for various diseases [88]. Computational methods such as docking, virtual screening, simulations, and artificial intelligence/machine learning can be effectively used for drug repurposing and for faster drug development at affordable costs.

Network analysis of the HIV-1 PR subtype C using computational tools is another potential future aspect in developing new inhibitors. This helps to model the interactions, evolutionary dynamics, and implications of PR towards drug resistance. Structural interaction networks such as the protein-protein interaction network, drug-protein interaction network, and genetic interaction network could help in deepening our understanding of the function of the PR in HIV replication and targeting effectively for therapeutic purposes.

## Figures and Tables

**Figure 1 viruses-17-00850-f001:**
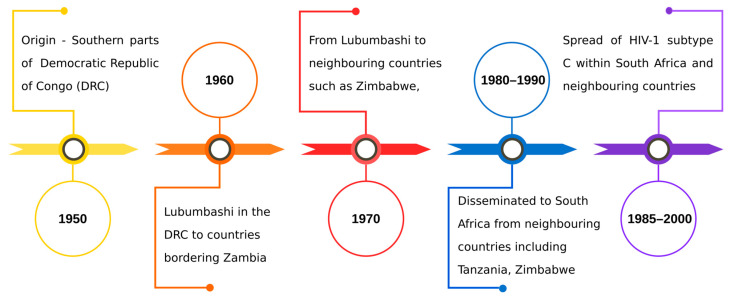
Origin and timeline of HIV-1 subtype C.

**Figure 2 viruses-17-00850-f002:**
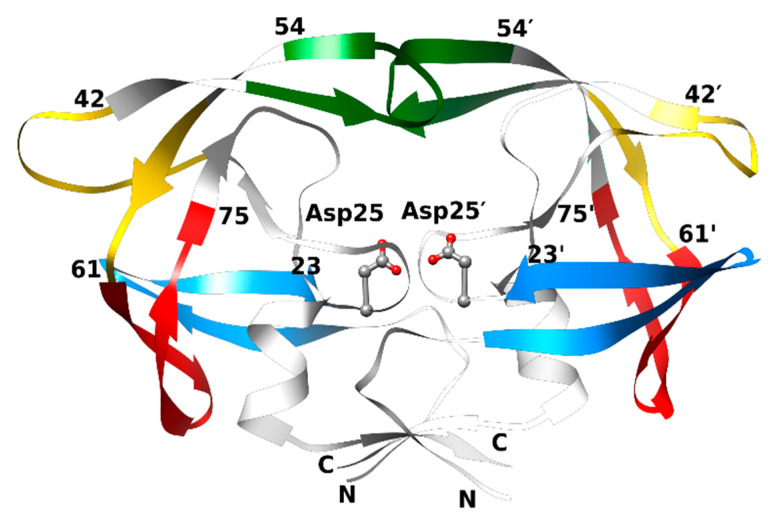
3D structure of HIV PR subtype C [37] highlighting functionally important regions such as the flaps (residues 46–54; green), hinge (residues 35–42 and 57–61; yellow), cantilever (residues 62–75; red) and fulcrum (residues 10–23; blue). The binding site residues Asp25/Asp25′ are shown in ball-and-stick representation. The figure was generated using the structure of HIV PR subtype B (PDB code: 2AQU) [38] as the template.

**Figure 3 viruses-17-00850-f003:**
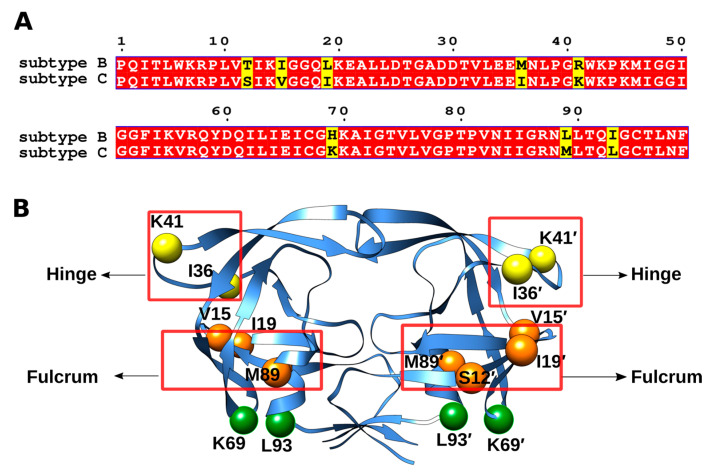
(**A**) Sequence alignment between subtype B and subtype C showing the positions with naturally occurring polymorphisms (yellow). (**B**) The residues associated with naturally occurring polymorphisms in HIV-1 PR subtype C (S12, V15, I19, I36, K41, K69, M89 and L93) are represented as spheres. The colors yellow, orange, and green correspond to the functional regions of the hinge, fulcrum, and the base of the protease, respectively. The figure was generated using the structure of HIV PR subtype B (PDB code: 2AQU) [38] as the template.

**Figure 4 viruses-17-00850-f004:**
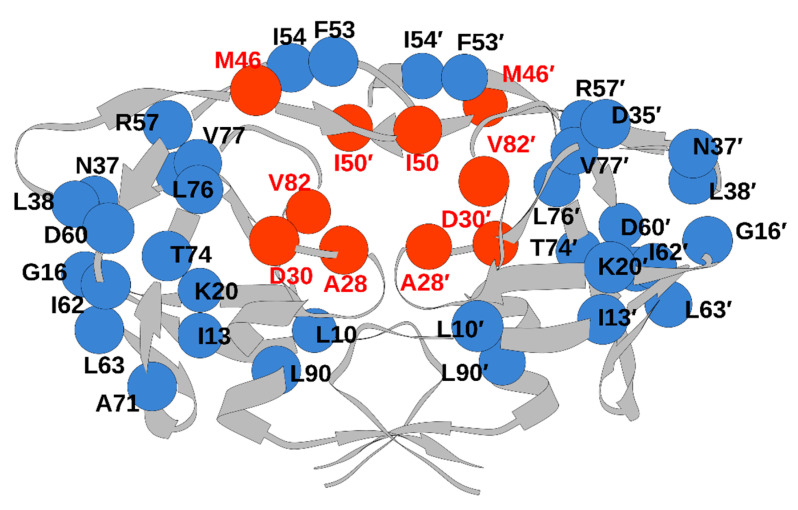
Residues in HIV-1 PR subtype C that are prone to mutations (blue: away from the active site; red: active site).

**Figure 5 viruses-17-00850-f005:**
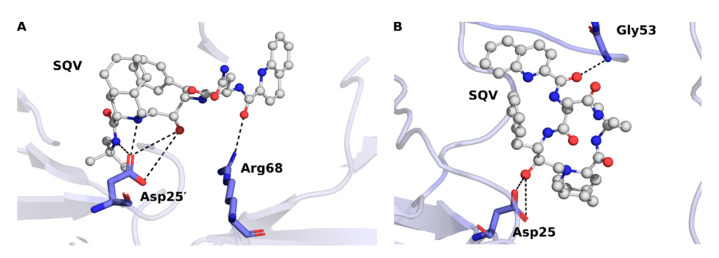
The hydrogen bond interactions of HIV-1 PR subtype C with SQV (**A**) WT and (**B**) L38HL. The figure was adapted from Venkatachalam et al. [59].

**Figure 6 viruses-17-00850-f006:**
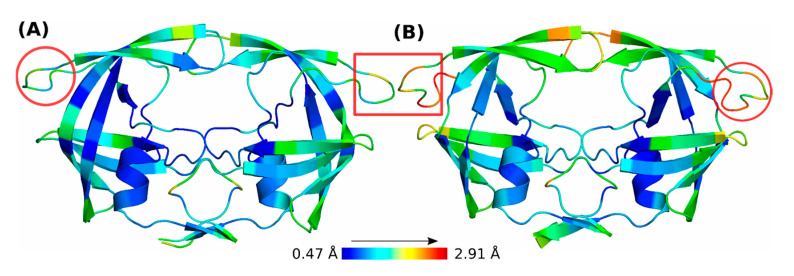
Comparison of hinge regions (red circles and rectangles) showing lower (blue) and higher (red) fluctuations in (**A**) WT and (**B**) L38HL HIV-1 PR subtype C. The figure was adapted from Sankaran et al. [37].

**Table 1 viruses-17-00850-t001:** List of drug-resistant substitutions observed in HIV-1 PR subtype C.

Mutation	Drug	Amino Acid Substitutions	References
Primary substitutions	NFV	D30N, M46I	[45]
	ATV	V82I	[41]
Secondary substitutions	ATV	L10I, G16E, K20R, M36I/V/L, D60E, I64V/L, L63P/T, T74S/A, I93L	[45]
	IDV	L10I, G16E, K20R, M36I, I62V, T74S, V77I	
	LPV	L10I, K20R, L63P, I64V	
	NFV	L10I, K20R, M36I, I62V, I64V, T74S/A, V77I	
	SQV	L10I, k20R, I62V, T74S/A, V77I	
	FPV	K20R	
	TPV	M36I/V/L, H69K, L89M	

NFV: Nelfinavir; ATV: Atazanavir; IDV: Indinavir; LPV: Lopinavir; SQV: Saquinavir; FPV: Fosamprenavir; TPV: Tipranavir.

**Table 2 viruses-17-00850-t002:** List of other substitutions observed/reported in HIV-1 PR subtype C.

Substitution	Amino Acid Change	References
Primary	V28A, I50L	[39,40]
Secondary	I13V, L33F, E35D, F53L, I54V, R57K, L63P/V/S/A/T, I64M, A71V, L76V, L90M	

**Table 3 viruses-17-00850-t003:** List of insertions observed in HIV-1 PR subtype C.

Insertions	Amino acid Change	Associated Mutations	Reference
Single insertions	E35↑E/Q, N37T↑V/N	None	[40]
	E35↑T	K20T	
Double insertions	D35↑G↑S	M46L	
	L38↑N↑L, L38↑H↑L	None	
	E35D↑G↑S	E35D, D60E	[49]
	I36↑T↑T	P39S, D60E and Q61E	[25]

**Table 4 viruses-17-00850-t004:** Binding free energy (ΔG) values of four FDA-approved drugs against subtype B and subtype C.

Drug	Binding Free Energy (ΔG) kcal/mol	
	WT Subtype B	WT Subtype C
SQV	−12.80	−12.40
RTV	−14.40	−13.40
IDV	−12.70	−12.30
NFV	−13.10	−13.00

SQV: Saquinavir; Ritonavir (RTV); IDV: Indinavir; FPV: Fosamprenavir; NFV: Nelfinavir. Data are taken from [23].

## Data Availability

No new data were created.

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
