# Peer review of "Integrative Computational Approaches for Understanding Drug Resistance in HIV-1 Protease Subtype C"

_viruses, 2025, doi:10.3390/v17060850_

Round 1

Reviewer 1 Report

Comments and Suggestions for Authors

The genetic diversity, mutations, and use of computational studies, especially Molecular Dynamics (MD) simulations, to comprehend drug resistance in HIV-1 Protease (PR) Subtype C are highlighted in this review article by Sankaran Venkatachalam et al. Future directions and existing research gaps are also covered. Given that subtype C is a major global health concern and that the majority of currently available protease inhibitors (PIs) are made to target subtype B, the manuscript tackles a topic that is both highly relevant and crucial to HIV research. The review integrates experimental and computational findings to give a thorough overview of current knowledge regarding drug resistance in HIV-1 PR subtype C. It is well-structured. This timely and thorough researched review significantly advances our knowledge of HIV-1 Protease Subtype C drug resistance. The use of computational approaches is a noteworthy strength, and the manuscript is well-structured. To address the aforementioned points, I suggest making minor revisions. The manuscript needs minor revision. 
Given the global prevalence of HIV-1 PR subtype C and the difficulties it presents for currently available antiretroviral treatments, the topic is pertinent and timely. The origin, genetic diversity, naturally occurring polymorphisms, drug-resistant mutations (substitutions and insertions), and extensive computational studies on both first- and second-generation PIs against HIV-1 PR subtype C are all methodically covered in the review's thorough coverage. The manuscript effectively combines computer methods, especially MD simulations and binding free energy calculations, to help us understand how drug resistance works at a molecular level and to find key interacting parts. The logical presentation of the content, with distinct headings and subheadings, guides the reader through complex information. Figures 1-5 clearly show key topics such as the history and origin of HIV-1 subtype C, the 3D shape of HIV PR, sequence alignments and variable sites, hydrogen bond interactions, and changes in the hinge region. Tables 1, 2, and 3 provide useful summaries of drug-resistant substitutions, other reported substitutions, and insertions, respectively. The FDA-approved drugs' binding free energy values against subtypes B and C are compiled in Table 4. The manuscript highlights the necessity of creating drugs that specifically target subtype C while skillfully identifying research gaps and future directions. 

1. abstract is instructive; it could be a little bit shorter without losing important details. 
2. While the introduction is well-written, there are a few paragraph transitions that could be smoother to enhance readability. 
3. Make sure all terms are used consistently in the manuscript. For instance, the terms "polymorphisms" and "mutations" are used, and although they are different, it might be helpful to clarify their relationship in the context of drug resistance at specific points. 
4. The caption of Figure 3 describes the colors for the hinge, fulcrum, and base, which are the functional regions. It would be helpful to clearly label the corresponding regions in the diagram or include a brief legend within the figure for easy reference, even though the image displays yellow, orange, and green spheres. 
5. Ensure that the main text mentions each figure and table logically, sequentially, and with explicit references. For instance, Figure 3 on Page 5 does not depict the location of drug-resistant substitutions, despite the text on Page 6 mentioning it. Rather, it displays polymorphic sites. For this purpose, there may be a minor discrepancy or a need for a new figure. The fact that the figures on pages 5 and 6 are labeled as the same number, figure 3, could be the cause. It is recommended that the explanation be rewritten in the text and that the quantity of figures displayed on pages 5 and 6 be adjusted. 
6. In the footnote, list the complete names of the drugs that are abbreviated in Table 1. 
7. In reference to drug-resistant substitutions from Table 1, the text states that "their respective locations on the HIV-1 PR subtype C structure are shown in Figure 3" (lines 170–171). On the other hand, naturally occurring polymorphisms are depicted in Figure 3B. Please either make a new figure that explicitly shows the locations of drug-resistant substitutions from Table 1 or update Figure 3B to include drug-resistant substitutions as well.

Author Response

Reviewer #1

The genetic diversity, mutations, and use of computational studies, especially Molecular Dynamics (MD) simulations, to comprehend drug resistance in HIV-1 Protease (PR) Subtype C are highlighted in this review article by Sankaran Venkatachalam et al. Future directions and existing research gaps are also covered. Given that subtype C is a major global health concern and that the majority of currently available protease inhibitors (PIs) are made to target subtype B, the manuscript tackles a topic that is both highly relevant and crucial to HIV research. The review integrates experimental and computational findings to give a thorough overview of current knowledge regarding drug resistance in HIV-1 PR subtype C. It is well-structured. This timely and thorough researched review significantly advances our knowledge of HIV-1 Protease Subtype C drug resistance. The use of computational approaches is a noteworthy strength, and the manuscript is well-structured. To address the aforementioned points, I suggest making minor revisions. The manuscript needs minor revision.

Given the global prevalence of HIV-1 PR subtype C and the difficulties it presents for currently available antiretroviral treatments, the topic is pertinent and timely. The origin, genetic diversity, naturally occurring polymorphisms, drug-resistant mutations (substitutions and insertions), and extensive computational studies on both first- and second-generation PIs against HIV-1 PR subtype C are all methodically covered in the review's thorough coverage. The manuscript effectively combines computer methods, especially MD simulations and binding free energy calculations, to help us understand how drug resistance works at a molecular level and to find key interacting parts. The logical presentation of the content, with distinct headings and subheadings, guides the reader through complex information. Figures 1-5 clearly show key topics such as the history and origin of HIV-1 subtype C, the 3D shape of HIV PR, sequence alignments and variable sites, hydrogen bond interactions, and changes in the hinge region. Tables 1, 2, and 3 provide useful summaries of drug-resistant substitutions, other reported substitutions, and insertions, respectively. The FDA-approved drugs' binding free energy values against subtypes B and C are compiled in Table 4. The manuscript highlights the necessity of creating drugs that specifically target subtype C while skillfully identifying research gaps and future directions.

Response: We sincerely thank the reviewer for reviewing the manuscript and for providing valuable comments to improve it further.

Comments

  1. abstract is instructive; it could be a little bit shorter without losing important details.

Response: We thank the reviewer for the thoughtful suggestion. The abstract is shortened without losing important details.

  1. While the introduction is well-written, there are a few paragraph transitions that could be smoother to enhance readability.

Response: We thank the reviewer for the suggestion. In response to that, we have made transitions smooth from paragraph to paragraph for better readability.

  1. Make sure all terms are used consistently in the manuscript. For instance, the terms "polymorphisms" and "mutations" are used, and although they are different, it might be helpful to clarify their relationship in the context of drug resistance at specific points.

Response: We sincerely thank the reviewer for pointing this out. We have made sure that the all the terms are used consistently in the manuscript. In addition, we provided the definitions for polymorphisms and drug-resistant mutations (line no. 143).

  1. The caption of Figure 3 describes the colors for the hinge, fulcrum, and base, which are the functional regions. It would be helpful to clearly label the corresponding regions in the diagram or include a brief legend within the figure for easy reference, even though the image displays yellow, orange, and green spheres.

Response: We thank the reviewer for the suggestion. We have now labelled the hinge, and the fulcrum regions of HIV PR subtype C in the Figure 3B for easy understanding.

  1. Ensure that the main text mentions each figure and table logically, sequentially, and with explicit references. For instance, Figure 3 on Page 5 does not depict the location of drug-resistant substitutions, despite the text on Page 6 mentioning it. Rather, it displays polymorphic sites. For this purpose, there may be a minor discrepancy or a need for a new figure. The fact that the figures on pages 5 and 6 are labeled as the same number, figure 3, could be the cause. It is recommended that the explanation be rewritten in the text and that the quantity of figures displayed on pages 5 and 6 be adjusted.

Response: We sincerely thank the reviewers for pointing this out. The label for Figure 3 in page 7 has been changed to Figure 4 (line no. 203). Accordingly, references corresponding to Figure 4 were corrected in the manuscript. Furthermore, subsequent figure labels were changed and their corresponding references were also updated.

  1. In the footnote, list the complete names of the drugs that are abbreviated in Table 1.

Response: We thank the reviewer for the for the comment. The complete names of the drugs are included in footnote in Table 1 (line no. 194).

  1. In reference to drug-resistant substitutions from Table 1, the text states that "their respective locations on the HIV-1 PR subtype C structure are shown in Figure 3" (lines 170–171). On the other hand, naturally occurring polymorphisms are depicted in Figure 3B. Please either make a new figure that explicitly shows the locations of drug-resistant substitutions from Table 1 or update Figure 3B to include drug-resistant substitutions as well.

Response: We sincerely thank the reviewers for pointing this out. The discrepancy was due to the ambiguous figure labels. The label for Figure 3 in page 7 has been changed to Figure 4 (line no. 203) and the text “their respective locations on the HIV-1 PR subtype C structure are shown in Figure 3” has been changed to “their respective locations on the HIV-1 PR subtype C structure are shown in Figure 4” (line no. 188-189).

Reviewer 2 Report

Comments and Suggestions for Authors

This is a review of the manuscript entitled, “Integrative Computational Approaches for Understanding Drug Resistance in HIV-1 Protease Subtype C”, by Sankaran Venkatachalam, Nisha Muralidharan, Ramesh Pandian, Yasien Sayed and 4 M. Michael Gromiha.

The main point made by the authors is that HIV-1 protease subtype C is naturally resistant to PIs that were designed for HIV-1 protease subtype B. And that efforts should be directed at designing PIs specific for the HIV-1 protease subtype C. The authors then review the current state of PI testing and development, along with strategies that could be used to develop PIs that target HIV-1 protease subtype C.

Main Comments

  1. I) In 2012 “only’ 30% of HIV-1 infected individuals in the US and Puerto Rico were on effective therapy (1). And since then, new combination HIV therapies have been approved by the FDA that do not include PIs, like VOCABRIA, CABENUVA, and DOVATO. This suggests that HIV-1 subtypes in the US evolved PI resistance since in the 1990s PIs, in combination with RT inhibitors, were key in controlling HIV-1 infections in the US. So the question then is why haven’t US pharmaceutical companies tried to make new PIs that target PI resistant HIV-1 subtype B strains? It would be informative to include a discussion of this in the manuscript, since as the authors state, HIV-1 subtype C expresses a PI resistant PR. 

    In addition, the authors stated on line 88 that “all the approved inhibitors were designed to target subtype B PRs”, there needs to be references to support that statement.

  2. II) It has been reported that HIV-1 PR can bind additional cleavage-site residues, outside of the cleavage site residues P4-P4’ that are bound by the active site, using the Substrate-grooves (2). Independent experimental support for the Substrate-grooves has been published (3). In this manuscript’s Fig. 3, Table 1 and 2, and the second Fig. 3, many of the mutations shown are in the Substrate-grooves. Selected Substrate-groove residues have been shown to make direct H-bonds to Gag cleavage site residues (2). This mechanism can allow Gag cleavage sites to outcompete Pi binding to the HIV PR. It may be helpful to identify HIV-1 PR subtype C residues in the Substrate-grooves that have been reported to bind to Gag cleavage sites. And that is an additional strategy to purse in PI development, PIs that bind to the PR Substrate-grooves.

III) In addition to the PI resistance mechanisms described in the manuscript, Gag non-cleavage-site PI resistance mutations are another way in which HIV-1 can be resistant to PIs (4). It was reported a Gag non-cleavage-site PI resistance mutation (H219Q) increased over time from 1997-2018 in the general population of treatment naive HIV-1 infected individuals (5). Have the authors looked to see if HIV-1 subtype C contains Gag non-cleavage-site PI resistance mutations? The HIV Gag non-cleavage-site PI resistant mutations were reported to stabilize the Gag Substrate-clamps (5), allowing Gag cleavage sites to out compete PI binding to the PR active site. Experimental support for the Substrate-clamps was found by reinterpreting previously published work (6, 7). Targeting the Substrate-clamps with inhibitors would be another strategy to inhibit the PR, since inhibitors targeting the Substrate-clamps could allow Gag cleavage sites to out compete PIs, while also potentially disrupting the ordered cleavage of Gag resulting in non-infectious virus.

Line-by-line comments

  • Line 14, The vast knowledge and widespread availability,

Comment Please delete “vast knowledge and”, it does not fit in the context of the sentence.

  • Line 15-16, There are ten FDA-15 approved protease inhibitors (PIs), which serve as the first-line defense in HIV treatment.

Comment Please note that VOCABRIA, CABENUVA, and DOVATO are current first-line HIV therapies approved by the FDA and available in the USA, however, none of them contain PIs.

  • Line 32, one of the most devastating diseases in human history [1-2].

Comment It is a serious disease, but not as severe as the black plague or small pox in a naive population, for example.

  • Line 38, ---allowing the virus to inject its genetic

Comment Please add to the end of the sentence “into the host cell”.

  • Line 68, --trend that subtype C is accounting for ~46 % of HIV infections worldwide [20-21].

Comment Please insert/correct as indicated: trend”, and” that subtype C “accounts” for ~46 % of HIV infections worldwide [20-21].

  • Line 77-78, key enzymes such as RT, PR and IN are primarily designed for subtype B, rendering them ineffective when used for subtype C.

Comment This statement needs a reference to support it.

  • Line 78-79, RT and IN 78 play vital roles in various stages, including viral fusion and viral-host DNA integration,

Comment How is RT involved in viral fusion?

  • Figure 2 line 110

Comment There needs to be a PDB identifier for the structure, and a reference for the authors who published the structure “in the figure legend”.

  • Figure 3 line 137

Comment There needs to be a PDB identifier for the structure, and a reference for the authors who published the structure “in the figure legend”.

  • Line 166-167, Further, patients who 166 are nonresponsive to--

Comment Please correct to: “Furthermore”, and replace are with “were”. I would recommend replacing all instances of Further with Furthermore.

  • Figure 3 Line 179-180. Residues in HIV-1 PR subtype C that are prone to mutations (blue: away from the active site; red: active site).

Comment Please note Figure 3 has already been used (Line 137). Change Fig. 3 to “Fig. 4”, and correct all references to the Figure in the text.

  • Line 200-202, While differences between subtype B and C are subtle, it was shown that the presence of mutations will decrease the interaction strength of the drugs to PR, culminating in drug [22, 38].

Comment What mutations are you referring to, can the mutant PRs and the binding free energy be added to Table 4? Please fill in a missing term “culminating in drug _____”.

  • Line 226-227, ---CD4, a key viral progression receptor,

Comment Please clarify what is CD4, and how the virus interacts with CD4.

  • Figure 4 line 272. It would be helpful to include the “hydrogens atoms” for the PR residues, and drug atoms, that interact by Hydrogen bonds.

Comment Please change Figure 4 to “Figure 5”, and correct all references to the Figure in the text.

  • Line 309, ---like valine or leucine affects the hydrophobic--

Comment Please change to past tense, “affected”. And check the rest of the manuscript for the same instances of using present tense.

  • Line 311-314, Sankaran et al., [54] reported that the insertion in the L38HL variant of HIV-1 PR increases the flexibility at the hinge regions leading to a unique binding mechanism for ATV by creating intramolecular hydrogen bonding in the ligand. This restricted the conformational flexibility and promoted a compact structure of the ligand (Figure 5).

Comment The text describes ATV, an inhibitor, but there is no inhibitor or ligand present in Fig. 5 (to be changed to Figure 6, see next comment).

  • Figure 5 line 327

Comment Change to “Figure 6” and correct all references to the Figure in the text.

  • Line 333-334, They observed that these mutations were possessed a considerable effect on the binding affinities of PIsto the PR.

Comment Please delete “were”, and insert space between “PIs to”

  • Line 378-379, But tailoring drugs/inhibitors specifically for HIV-1 PR subtype C can inhibit HIV more effectively.

Comment Please correct: “By” tailoring drugs/inhibitors specifically for HIV-1 PR subtype C “they could” inhibit HIV more effectively.

  • Line 399, --- Stanford HIV database (HIVDB).

Comment Please include a reference with the website address.

  • Line 401-402, This DIF model in the presence of an isolate is effective—

Comment Please clarify “an isolate”

References

  • Frieden, T.R.; Foti, K.E.; Mermin, J. Applying Public Health Principles to the HIV Epidemic—How Are We Doing? N. Engl. J. Med. 2015, 373, 2281–2287.
  • Laco, G.S. HIV-1 protease substrate-groove: Role in substrate recognition and inhibitor resistance. Biochimie 2015, 118, 90–103.
  • Miczi, M.; Diós, Á.; Bozóki, B.; Tozsér, J.; Mótyán, J. Development of a Bio-Layer Interferometry-Based Protease Assay Using HIV-1 Protease as a Model. Viruses 2021, 13, 1183.
  • Su,C.T.-T.; Koh, D.W.-S.; Gan, S.K.-E. Reviewing HIV-1 Gag Mutations in Protease Inhibitors Resistance: Insights for Possible Novel Gag Inhibitor Designs. Molecules 2019, 24, 3243.
  • Laco, G.S. HIV-1 Gag Non-Cleavage Site PI Resistance Mutations Stabilize Protease/Gag Substrate Complexes in Silico via a Substrate-Clamp. BioChem 2021, 1, 190–209.
  • Pettit, S.; Moody, M.D.; Wehbie, R.S.; Kaplan, A.H.; Nantermet, P.V.; Klein, C.A.; Swanstrom, R. The p2 domain of human immunodeficiency virus type 1 Gag regulates sequential proteolytic processing and is required to produce fully infectious virions. J. Virol. 1994, 68, 8017–8027.
  • Deshmukh, L.; Louis, J.M.; Ghirlando, R.; Clore, G.M. Transient HIV-1 Gag–protease interactions revealed by paramagnetic NMR suggest origins of compensatory drug resistance mutations. Proc. Natl. Acad. Sci. USA 2016, 113, 12456–12461.
Comments on the Quality of English Language

Overall the quality of the English was pretty good, with exceptions noted in the line-by-Line comments section.

Author Response

Reviewer 2

This is a review of the manuscript entitled,Integrative Computational Approaches for Understanding Drug Resistance in HIV-1 Protease Subtype C”, by Sankaran Venkatachalam, Nisha Muralidharan, Ramesh Pandian, Yasien Sayed and 4 M. Michael Gromiha

The main point made by the authors is that HIV-1 protease subtype C is naturally resistant to PIs that were designed for HIV-1 protease subtype B. And that efforts should be directed at designing PIs specific for the HIV-1 protease subtype C. The authors then review the current state of PI testing and development, along with strategies that could be used to develop PIs that target HIV-1 protease subtype C.

Response: We sincerely thank the reviewer for reviewing the manuscript and for providing valuable comments to improve it further.

Main Comments

  1. In 2012 “only’ 30% of HIV-1 infected individuals in the US and Puerto Rico were on effective therapy (1). And since then, new combination HIV therapies have been approved by the FDA that do not include PIs, like VOCABRIA, CABENUVA, and DOVATO. This suggests that HIV-1 subtypes in the US evolved PI resistance since in the 1990s PIs, in combination with RT inhibitors, were key in controlling HIV-1 infections in the US. So the question then is why haven’t US pharmaceutical companies tried to make new PIs that target PI resistant HIV-1 subtype B strains? It would be informative to include a discussion of this in the manuscript, since as the authors state, HIV-1 subtype C expresses a PI resistant PR.

Response: We greatly appreciate the thoughtful observation of the reviewer regarding the evolution of PI resistance and the shifting landscape of antiretroviral therapy in the United States. The reason that pharmaceutical companies focused attention on the development of novel therapies, VOCABRIA, DOVATO and CABENUVA, that purposefully do not include PIs so as to circumvent the challenge/issue of designing new PI drugs against a range of proteases incorporating multi-drug resistant mutations already in circulation. We have now included this discussion in the manuscript (lines 93-103).

  1. In addition, the authors stated on line 88 that “all the approved inhibitors were designed to target subtype B PRs”, there needs to be references to support that statement.

Response: We thank the reviewer for pointing out the need to support our statement regarding the development of PIs and their subtype specificity. In response to that, we have added references in the manuscript (line no. 93).

  1. It has been reported that HIV-1 PR can bind additional cleavage-site residues, outside of the cleavage site residues P4-P4’ that are bound by the active site, using the Substrate-grooves (2). Independent experimental support for the Substrate-grooves has been published (3). In this manuscript’s Fig. 3, Table 1 and 2, and the second Fig. 3, many of the mutations shown are in the Substrate-grooves. Selected Substrate-groove residues have been shown to make direct H-bonds to Gag cleavage site residues (2). This mechanism can allow Gag cleavage sites to outcompete Pi binding to the HIV PR. It may be helpful to identify HIV-1 PR subtype C residues in the Substrate-grooves that have been reported to bind to Gag cleavage sites. And that is an additional strategy to purse in PI development, PIs that bind to the PR Substrate-grooves.

Response: We sincerely thank the reviewer for the thoughtful suggestion about the role of the substrate-groove in HIV-1 protease and its potential relevance to Gag cleavage site binding and PI development. We have included the details in the revised manuscript (lines 205-208 & 459-465).

  1. In addition to the PI resistance mechanisms described in the manuscript, Gag non-cleavage-site PI resistance mutations are another way in which HIV-1 can be resistant to PIs (4). It was reported a Gag non-cleavage-site PI resistance mutation (H219Q) increased over time from 1997-2018 in the general population of treatment naive HIV-1 infected individuals (5). Have the authors looked to see if HIV-1 subtype C contains Gag non-cleavage-site PI resistance mutations? The HIV Gag non-cleavage-site PI resistant mutations were reported to stabilize the Gag Substrate-clamps (5), allowing Gag cleavage sites to out compete PI binding to the PR active site. Experimental support for the Substrate-clamps was found by reinterpreting previously published work (6, 7). Targeting the Substrate-clamps with inhibitors would be another strategy to inhibit the PR, since inhibitors targeting the Substrate-clamps could allow Gag cleavage sites to out compete PIs, while also potentially disrupting the ordered cleavage of Gag resulting in non-infectious virus.

Response: We thank the reviewer for raising this important point regarding non-cleavage-site mutations in Gag and their potential contribution to PI resistance, particularly in HIV-1 subtype C. We have included the details in Section 6 with relevant references. (lines 435-447).

Line-by-line comments

  1. Line 14, The vast knowledge and widespread availability,

Comment: Please delete “vast knowledge and”, it does not fit in the context of the sentence.

Response: We thank the reviewer for the suggestion. The phrase “vast knowledge” has now been removed.

  1. Line 15-16, There are ten FDA-15 approved protease inhibitors (PIs), which serve as the first-line defense in HIV treatment.

Comment: Please note that VOCABRIA, CABENUVA, and DOVATO are current first-line HIV therapies approved by the FDA and available in the USA, however, none of them contain PIs.

Response: We agree with the reviewer that there are HIV therapies, which do not include PIs such as VOCABRIA, CABENUVA, and DOVATO. This information has been included in the revised manuscript (lines 98-99)

  1. Line 32, one of the most devastating diseases in human history [1-2]

Comment: It is a serious disease, but not as severe as the black plague or small pox in a naive population, for example.

Response: We thank the reviewer for pointing it out. The sentence has been changed to “one of the deadly diseases in human history” (line no. 31)

  1. Line 38, ---allowing the virus to inject its genetic

Comment: Please add to the end of the sentence “into the host cell”.

Response: As per the reviewer’s suggestion, the phrase “into the host cell” has been added to the end of the sentence (line no. 38).

  1. Line 68, --trend that subtype C is accounting for ~46 % of HIV infections worldwide [20-21].

Comment: Please insert/correct as indicated: trend”, and” that subtype C “accounts” for ~46 % of HIV infections worldwide [20-21].

Response: As per the reviewer’s suggestion, the sentence has been changed to “trend, and that subtype C accounts for ~46 % of HIV infections worldwide” (line no. 69)

  1. Line 77-78, key enzymes such as RT, PR and IN are primarily designed for subtype B, rendering them ineffective when used for subtype C.

Comment: This statement needs a reference to support it.

Response: We thank the reviewer for pointing it out. The references supporting the above statement has now been included in the manuscript (lines 82-83).

  1. Line 78-79, RT and IN 78 play vital roles in various stages, including viral fusion and viral-host DNA integration,

Comment: How is RT involved in viral fusion?

Response: We sincerely thank the reviewer for the question. RT is not involved in viral fusion. The sentence is now changed to “RT and IN play vital roles in various stages, including RNA to DNA conversion and viral-host DNA integration” in the manuscript (line no. 83).

  1. Figure 2 line 110

Comment There needs to be a PDB identifier for the structure, and a reference for the authors who published the structure “in the figure legend”.

Response: Thanks for the comment. Figure 2 is the computationally modelled 3D structure of HIV PR subtype C using 2AQU (HIV PR subtype B) as the template. We have provided references for both crystal and modelled structures in the revised manuscript (lines 127& 131).

  1. Figure 3 line 137

Comment There needs to be a PDB identifier for the structure, and a reference for the authors who published the structure “in the figure legend”.

Response: Thanks for the comment. Figure 2 is the computationally modelled 3D structure of HIV PR subtype C using 2AQU (HIV PR subtype B) as the template. We have provided references for both crystal and modelled structures in the revised manuscript (line no. 161).

  1. Line 166-167, Further, patients who 166 are nonresponsive to—

Comment: Please correct to: “Furthermore”, and replace are with “were”. I would recommend replacing all instances of Further with Furthermore.

Response: We thank the reviewer for the suggestion. As per the suggestion, the word “Further” has been replaced with “Furthermore” throughout the manuscript. Also, the word “are” is replaced with “were” (line no. 187).

  1. Figure 3 Line 179-180. Residues in HIV-1 PR subtype C that are prone to mutations (blue: away from the active site; red: active site).

Comment: Please note Figure 3 has already been used (Line 137). Change Fig. 3 to “Fig. 4”, and correct all references to the Figure in the text.

Response: Response: We sincerely thank the reviewers for pointing this out. The label for Figure 3 in page 7 has been changed to Figure 4 (line no. 203). Accordingly, references corresponding to Figure 4 were corrected in the manuscript. Furthermore, subsequent figure labels were changed and their corresponding references were also updated.

  1. Line 200-202, While differences between subtype B and C are subtle, it was shown that the presence of mutations will decrease the interaction strength of the drugs to PR, culminating in drug [22, 38].

Comment What mutations are you referring to, can the mutant PRs and the binding free energy be added to Table 4? Please fill in a missing term “culminating in drug _____”.

Response: We thank the reviewer for the question. We would like to clarify that the statement mentioned in the manuscript was from the cited paper. Nevertheless, we have now included one example supporting the statement (lines 230-233). Since we have provided an example of just one mutant, the binding free energies were not included in the table. Furthermore, the missing term is updated to “culminating in drug resistance” (line no. 233).

  1. Line 226-227, ---CD4, a key viral progression receptor,

Comment: Please clarify what is CD4, and how the virus interacts with CD4.

Response: We thank the reviewer for pointing out the need for greater clarity regarding CD4. We have included the details in the revised manuscript (lines 260-265).

  1. Figure 4 line 272. It would be helpful to include the “hydrogens atoms” for the PR residues, and drug atoms, that interact by Hydrogen bonds.

Comment: Please change Figure 4 to “Figure 5”, and correct all references to the Figure in the text.

Response: We sincerely thank the reviewer for pointing this out. The figure labels for the current and subsequent figures have been changed (Figure 4 to Figure 6) and changes are highlighted in the manuscript. We would like to clarify that the figure has been taken from the literature and hence, it is difficult to display hydrogen atoms of the PR residues.

  1. Line 309, ---like valine or leucine affects the hydrophobic—

Comment: Please change to past tense, “affected”. And check the rest of the manuscript for the same instances of using present tense.

Response: We thank the reviewer for this suggestion. The word “affects” has now been changed to “affected” (line no. 347). We have checked for similar instances in the manuscript and updated them.

  1. Line 311-314, Sankaran et al., [54] reported that the insertion in the L38HL variant of HIV-1 PR increases the flexibility at the hinge regions leading to a unique binding mechanism for ATV by creating intramolecular hydrogen bonding in the ligand. This restricted the conformational flexibility and promoted a compact structure of the ligand (Figure 5).

Comment: The text describes ATV, an inhibitor, but there is no inhibitor or ligand present in Fig. 5 (to be changed to Figure 6, see next comment).

Response: We sincerely thank the reviewer for pointing this out. Figure 6 was provided to illustrate the fact that the flexibility increases at the hinge region and corrected the sentence appropriately (lines 349-350).

  1. Figure 5 line 327

Comment: Change to “Figure 6” and correct all references to the Figure in the text.

Response: We sincerely thank the reviewer for pointing it out. The figure label corresponding to Figure 5 has been changed to Figure 6 (line no. 365) and their respective references are now updated in the manuscript.

  1. Line 333-334, They observed that these mutations were possessed a considerable effect on the binding affinities of PIsto the PR.

Comment: Please delete “were”, and insert space between “PIs to”

Response: Thank you for the suggestion and we incorporated the correction (line no. 371).

  1. Line 378-379, But tailoring drugs/inhibitors specifically for HIV-1 PR subtype C can inhibit HIV more effectively.

Comment Please correct: “By” tailoring drugs/inhibitors specifically for HIV-1 PR subtype C “they could” inhibit HIV more effectively.

Response: The line has been changed as per the suggestion of the reviewer (line no. 416-417).

  1. Line 399, --- Stanford HIV database (HIVDB).

Comment: Please include a reference with the website address.

Response: We thank the reviewer for pointing it out. We have now included the reference along with the website address in the manuscript (line no. 451).

  1. Line 401-402, This DIF model in the presence of an isolate is effective—

Comment: Please clarify “an isolate”

Response: Response: We thank the reviewer for pointing this out for clarification. The term "isolate" refers to a specific HIV-1 protease variant, typically defined by its unique amino acid sequence resulting from mutations, typically derived from patient samples.  To improve clarity, we have revised the sentence in the manuscript (lines 453-455).

Round 2

Reviewer 2 Report

Comments and Suggestions for Authors

Second review of “Integrative Computational Approaches for Understanding Drug Resistance in HIV-1 Protease Subtype C, Sankaran Venkatachalam , Nisha Muralidharan , Ramesh Pandian , Yasien Sayed , M. Michael Gromiha”.

Main Comment

Overall, a much-improved manuscript, good work authors. Please double check all new references to make sure the correct citation numbers are listed in the text.

Minor Comments

  • Line 58, Among these subtypes, understanding the distribution and prevalence of HIV-1 is essential.

Comment: This may read clearer; Understanding the distribution and prevalence of HIV-1 “subtypes is essential for effective treatment”.

  • Line 69, ---and that subtype C accounts for ~46 % of HIV infections worldwide.

Comment: and that subtype C “now” accounts for ~46 % of HIV infections worldwide.

  • Line 445, cited references [79–81] are not complete.

Comment: The complete references are [“78”-81]

  • Line 462, cited references [85,86] are not the correct references.

Comment: The correct references are [“46, 47”].

  • Line 462, Studies linking groove residues to altered---

Comment: this would be clearer for readers; Studies linking “substrate-“groove residues to altered---

Author Response

We thank the reviewer for the constructive comments. We have carefully addressed all the points and the corrections are included in the revised manuscript.

Main Comment

Overall, a much-improved manuscript, good work authors. Please double check all new references to make sure the correct citation numbers are listed in the text.

Response: We sincerely thank the reviewer for reviewing the manuscript and for providing valuable comments to improve it further. We have checked all new references and made sure that all references are correctly cited in the list and in the text.

Minor Comments

  1. Line 58, Among these subtypes, understanding the distribution and prevalence of HIV-1 is essential. Comment: This may read clearer; Understanding the distribution and prevalence of HIV-1 “subtypes is essential for effective treatment”.

Response: We thank the reviewer for the suggestion. We have incorporated the correction (line no. 59).

  1. Line 69, ---and that subtype C accounts for ~46 % of HIV infections worldwide.

Comment: and that subtype C “now” accounts for ~46 % of HIV infections worldwide.

Response: We have updated the statement accordingly (line no. 69).

  1. Line 445, cited references [79–81] are not complete. Comment: The complete references are [“78”-81]

Response: We have updated the references in the revised manuscript (line no. 444).

  1. Line 462, cited references [85,86] are not the correct references.

Comment: The correct references are [“46, 47”].

Response: There references are corrected in the revised manuscript (line no. 461).

  1. Line 462, Studies linking groove residues to altered---

Comment: this would be clearer for readers; Studies linking “substrate-“groove residues to altered---

Response: We have updated the statement (line no. 461)
